# Pr and Pfr structures of plant phytochrome A

Soshichiro Nagano [1,2,3,10] ✉, David von Stetten [4,10], Kaoling Guan [1,5,10], Peng-Yuan Chen [1], Chen Song [6], Thomas Barends[3], Manfred S. Weiss [7], Christian G. Feiler[7], Katerina Dörner [8], Iñaki de Diego Martinez [8], Robin Schubert [8], Johan Bielecki [8], Lea Brings[8], Huijong Han [8], Konstantin Kharitonov[8], Chan Kim [8], Marco Kloos [8], Jayanath C. P. Koliyadu [8], Faisal H. M. Koua [8], Ekaterina Round[8], Abhisakh Sarma [8], Tokushi Sato [8], Christina Schmidt [8], Joana Valerio [8], Agnieszka Wrona[8], Joachim Schulz [8], Raphael de Wijn [8], Romain Letrun [8], Richard Bean [8], Adrian Mancuso [8,9], Karsten Heyne[2] & Jon Hughes [1,2] ✉

Phytochromes are biliprotein photoreceptors widespread amongst micro-organisms and ubiquitous in plants where they control developmental processes as diverse as germination, stem elongation and floral induction through the photoconversion of inactive Pr to the Pfr signalling state. Here we report crystal structures of the chromophore-binding module of soybean phytochrome A, including ~2.2 Å XFEL structures of Pr and Pfr at ambient temperature and high resolution cryogenic structures of Pr. In the Pfr structure, the chromophore is exposed to the medium, the D-ring remaining α-facial following the likely clockwise photoflip. The chromophore shifts within its pocket, while its propionate side chains, their partners as well as three neighbouring tyrosines shift radically. Helices near the chromophore show substantial shifts that might represent components of the light signal. These changes reflect those in bacteriophytochromes despite their quite different signalling mechanisms, implying that fundamental aspects of phytochrome photoactivation have been repurposed for photoregulation in the eukaryotic plant.

Photosynthesis is the sole source of energy for plants and the only significant one for life on Earth. Consequently, the central environmental factor in regulating plant biochemistry and development is light. This regulation is achieved through sensory photoreceptors such as phototropins, cryptochromes and phytochromes. Alongside shade avoidance and flowering time, phytochromes mediate light-induced germination and seedling photomorphogenesis, both primarily through the action of phytochrome A (phyA) under natural conditions. Indeed, in *Arabidopsis*, phyA alone has major effects on the

transcription of at least 10% of all genes[1–3]. The principal role of phyA is to detect exceedingly low light levels by integrating photons over prolonged periods, whereby molecules in the inactive Pr state are converted to and accumulate as the Pfr signalling state in the nucleus (the very low fluence response or VLFR)[4,5]. Classical, red/far-red reversible photoresponses mediated by phyB are at least a thousand-fold less sensitive to light. phyA also mediates the high irradiance response (HIR) in strong far-red light (see ref. 6), although it is doubtful whether this can occur under natural conditions. How the Pfr structure

[1]Institute for Plant Physiology, Justus Liebig University, Giessen, Germany. [2]Department of Physics, Free University of Berlin, Berlin, Germany. [3]Department of Biomolecular Mechanisms, Max Planck Institute for Medical Research, Heidelberg, Germany. [4]European Molecular Biology Laboratory (EMBL), Hamburg, Germany. [5]Department of Molecular Biosciences, University of Texas at Austin, Austin, Texas, USA. [6]Department of Analytical Chemistry, University of Leipzig, Leipzig, Germany. [7]Helmholtz-Zentrum Berlin für Materialien und Energie, BESSY II, Macromolecular Crystallography, Berlin, Germany. [8]European XFEL GmbH, Schenefeld, Germany. [9]Diamond light source, Didcot, United Kingdom. [10]These authors contributed equally: Soshichiro Nagano, David von Stetten, Kaoling Guan. ✉e-mail: soshichiro.nagano@mr.mpg.de; jon.hughes@uni-giessen.de

differs from that of Pr, how Pr→Pfr photoactivation is brought about and how the light signal is then transmitted to the cell are thus important questions in biology (reviewed in ref. 7).

X-ray crystallography has shown that the 3D structure of the N-terminal photosensory module (PSM, see Supplementary Fig. 1 for domain map and domain definitions) of plant phytochromes in the Pr state[8,9] resembles that of prokaryotic phytochromes[10,11]. As the C-terminal region also shows rather clear homology to prokaryotic histidine kinases[12–14], it was widely assumed that the full-length plant photoreceptor, despite the evolutionary insertion of a PAS domain repeat, would also resemble the head-to-head (parallel) dimer seen in prokaryotes. Cryo-EM studies of full-length phyA in the Pr state presented a very different picture, however, whereby the photosensory module together with the PAS repeat form a head-to-tail (antiparallel) dimeric platform bound together by the parallel-dimeric, histidine-kinase-related C-terminal domain, together forming a mushroom-like structure[15–18]. The cryo-EM structure of phyB as Pr is quite similar, although interactions between the platform and the stalk result in an overall form resembling a crooked mushroom[16]. Surprisingly, however, recent cryo-EM studies of phyB Pfr in complex with a signalling partner show the PSM in a head-to-head (parallel) dimeric configuration[19].

Physiological studies with transgenic plants have shown that, as a dimer in the nucleus, the PSM alone can generate components of the light signal[20–24], probably deriving from a direct effect of Pfr interaction with PIF transcription factors on their ability to bind DNA[25]. In contrast, prokaryotic phytochromes signal by physically de/activating histidine kinase or other enzymatic functions associated with the C-terminal region[14,26–30]. Interestingly, kinase activity in prokaryotic phytochromes is associated with the Pr state, whereas Pfr is the signalling state in plants. Finally, although Cph1 from the cyanobacterium *Synechocystis* 6803 attaches its phytobilin chromophore to a conserved Cys residue in the GAF domain just as in plants[11,13,14,31,32], bacteriophytochromes use instead a biliverdin chromophore attached via a longer linkage to a Cys near the N-terminus[10,27].

In view of these differences between plant and prokaryotic phytochromes, it is important to determine the structural changes associated with photoactivation and intramolecular signalling for both. Although a plant phyB Pfr structure has been reported recently[19], to date, the only structural information on the Pfr state of plant phyA is from MAS NMR[33–35]. Here we present several novel 3D structures of a minimal soybean phyA(nPAS-GAF) construct as Pr and Pfr. Although the causality is not understood, phytochromes usually require at least the nPAS, GAF and PHY domains for photoconversion of Pr to stable

Pfr, the tongue – a remarkable hairpin loop of the PHY domain that stretches back to contact the GAF domain – playing a central role. Remarkably, however, our construct is stably photochromic not only in solution[9] but also *in crystallo*. It is therefore likely that crystals of this molecule represent structures functionally similar to those in solution (however, we draw attention to Raman data that imply subtle differences between the final photoproducts in nPAS-GAF and in more complete constructs[9]). Here we describe a cryogenic single-crystal (MX) structure of soybean phyA(nPAS-GAF) Pr at 1.58 Å resolution, to our knowledge currently the highest resolution for a plant phytochrome. We also used serial femtosecond X-ray (SFX; see ref. 36 for review) diffraction data at *ca.* 2.2 Å resolution collected at ambient temperature from microcrystals pre-exposed to either red or far-red light at the European X-ray free-electron laser (EuXFEL) to derive the 3D structures of both Pr and Pfr states. By comparing the various datasets and structures, we identify the changes associated with photoactivation in this region of the phyA molecule and discuss these in relation to phyB and to signalling.

## Results

### Cryogenic Pr structures

We previously published a 3D structure of the phycocyanobilin (PCB) adduct of soybean phyA(nPAS-GAF) from 2.1 Å diffraction data collected at 100 K at the BESSY II synchrotron (PDB code 6TC7)[9]. Here we report an improved model at 1.58 Å (PDB code 8R44; see Fig. 1 and Supplementary Table 1). Although the ethyl group of chromophore ring D is directed predominantly α-facially relative to the plane of the B/C-rings, positive electron density is apparent on the β face too, indicating that both conformations coexist.

We compared the 8R44 MX structure with the near full-length cryo-EM structures of phyA[15–18] (Fig. 1c). Although significant variability between all the structures is apparent, the general pattern is consistent. 8R44 is thus probably a faithful representation of the phyA structure in the nPAS-GAF region.

Phytochromobilin (PΦB) rather than PCB is the native chromophore in plant phytochromes. Indeed, the different adducts show subtle differences: not only is $\lambda_{max}$ red-shifted on account of the PΦB D-ring vinyl side chain, in phyB, the rate of thermal reversion in vitro is faster for PΦB- than for PCB-adducts[37]. In the case of the present phyA construct, the thermal reversion half-lives of the PCB and PΦB adducts were 112 and 55 min, respectively (Supplementary Fig. 2). We solved the MX structure of the latter (PDB code 8R45; 1.86 Å resolution, Supplementary Table 1), finding it to be almost identical to that of the PCB adduct (Supplementary Figs. 3 and 4).

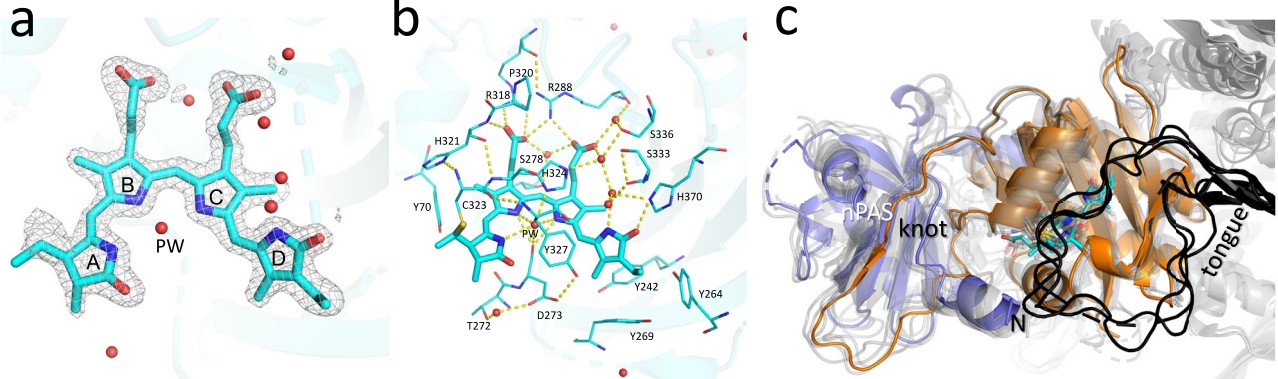

**Fig. 1 | High-resolution cryogenic crystal structure of soybean phyA(nPAS-GAF) as Pr.** The 1.58 Å cryogenic structure of the chromophore and pocket as Pr (PDB code 8R44, protomer B) is shown. Carbons, cyan. **a** 2Fo-Fc electron density map of the chromophore contoured at 0.314 e⁻ Å⁻³ (1.1 rmsd). **b** Amino acids and waters (red spheres) surrounding the chromophore. PW, pyrrole water. H-bonds are shown as dashed yellow lines. **c** 8R44 (nPAS domain, slate blue; GAF domain, orange; chromophore carbons, cyan; N, N-terminus) superimposed with published *ca.* 3.2 Å near full-length cryo-EM structures of phyA from *Arabidopsis* and maize (8F5Z, 8IFF, 8ISJ & 8ISK; PHY domain tongue, black; otherwise transparent grey), all as Pr.

## Ambient temperature Pr structures

We used the SPB/SFX (single particles, clusters and biomolecules / serial femtosecond crystallography) instrument at EuXFEL to determine the structure of soybean phyA(nPAS-GAF)-PCB at ambient temperature. Modifications of the crystallisation conditions, in particular seeding and batch processing rather than vapour diffusion, yielded plate-like microcrystals (1–10 µm in the longer axes, see Supplementary Discussion and Supplementary Fig. 5). Following saturating irradiation with far-red light (FR) from 730 nm LEDs, the microcrystal slurry was agitated gently in darkness prior to jetting using a double-flow focusing nozzle (DFFN, see Methods section) at ambient temperature (294 K). The Pr structure was determined and refined to 2.2 Å resolution on the basis of 61,275 diffraction images (PDB code 9ER4; Supplementary Table 1). Previously unresolved regions such as the 150s and 380s loops (chain A 111–118 and 346–359; chain B 111–120 and 346–359) remained unresolved, indicating that the molecular disorder is not an artefact of freezing and probably reflects the inherent mobility of those regions.

We compared 9ER4 with a replicate SFX dataset from EuXFEL collected independently, as well as with a 2.8 Å dataset from the T-REXX endstation at PETRA III (DESY / EMBL, Hamburg), also at ambient temperature (Supplementary Discussion and Supplementary Fig. 6). Fo-Fo electron density difference maps and difference distance matrix[38] analyses of the SFX data showed that the structures are highly reproducible. We similarly compared the 9ER4 and 8R44 structures to estimate differences related to cryogenic vs. ambient temperature conditions (Supplementary Discussion and Supplementary Fig. 7). The 244 K temperature difference gave rise to a 1.2% expansion of the crystallographic dimer, corresponding to a mean linear expansion coefficient of $49 \times 10^{-6}$ / K, whereas the unit cell dimensions (Supplementary Table 1) increased by 2.4% or $96 \times 10^{-6}$ / K. These values are similar to those of other protein crystals[39] and explain most of the differences between the MX and SFX structures: accordingly, local superimpositions show the structures to be almost identical (Supplementary Fig. 8).

## Ambient temperature Pfr structure

Although the PHY domain is required for stable Pfr formation in most plant and prokaryotic phytochromes, we found that our phyA(nPAS-GAF) construct could be photoconverted between Pr and Pfr not only in solution[9] but also *in crystallo*. Our early attempts to determine the Pfr structure using red-light-irradiated crystals (longer axes ~100 µm) were unsuccessful due to the light gradient resulting from the 16 mM protein concentration *in crystallo*: only ~1% of the actinic light would penetrate further than ~10 µm into the crystal, assuming an extinction coefficient of ~110 mM$^{-1}$cm$^{-1}$[40]. On the same basis, however, penetration of omnidirectional light into the thin, plate-like microcrystals would be adequate. Indeed, we measured strong photochromicity of the microcrystals washed and resuspended in protein-free precipitant (Supplementary Fig. 5d).

In order to collect diffraction data for Pfr at the XFEL, we thus irradiated the sample in the capillary that feeds the slurry to the jetting nozzle with a 630 nm LED array, allowing ~45 s for dark relaxation to the final photoproduct before the sample arrived at the X-ray focal spot. We then calculated the mixed-state electron density map from 176,334 microcrystal diffraction images (PDB 9QZT) and derived the Fo(light)-Fo(dark) difference map. Changes associated with photoactivation were readily apparent in the chromophore region (Supplementary Discussion and Supplementary Fig. 9), for example, around Y242 where a radically shifted side chain rotamer was expected according to MAS NMR of oat phyA3[33] and bacteriophytochrome crystal structures.

The Fo(light)-Fo(dark) map, as well as extrapolated maps[41] were used to derive a model of Pfr from the Pr and mixed-state SFX datasets (Supplementary Fig. 10). Paying particular regard to the Y242 rotamer, the Pfr occupancy was estimated to be ~11% and ~23% for the A- and B-protomers, respectively. It is unclear why the occupancies of the chains differ. The pure Pfr structure was deposited as PDB 9F4I (Supplementary Table S1) following the final refinement as a mixture of Pfr and Pr with different Pfr occupancies per chain (see Methods and Supplementary Figs. 9 and 10). We analysed the global changes associated with Pr→Pfr photoconversion using Fo(light)-Fo(dark) maps and difference distance matrices (Fig. 2). The largest electron density differences are associated with the chromophore region of chain B, consistent with the higher Pfr occupancy there. From the distance difference matrices, specifically chain A shows a slight general shift of the nPAS domain, whereas chain B shows slight shifts of residues K78-V110. Both chains show shifts corresponding to helices Q276-K286 and H321-D332 (blue and red stripes in Fig. 2b, respectively).

The chromophore region associated with the B-chain Pfr state is shown in Fig. 3. As in the case of Pr, electron density corresponding to the ethyl side chain is apparent on both sides of ring D. An α-facial disposition was nevertheless modelled on the basis of MAS NMR data for oat phyA3 Pfr[33]. Unfortunately, the Pfr electron density map does not allow for unambiguous determination of the ring-C propionate (propC) geometry.

## Discussion

### Pr structures

We present 1.58 and 1.86 Å cryogenic MX structures derived from single crystals of PCB and PΦB adducts (PDB codes 8R44 and 8R45, Fig. 1 and Supplementary Fig. 3, respectively; more detailed electron density and omit maps are shown in Supplementary Figs. 11 and 12) of soybean phyA(nPAS-GAF) as Pr, both broadly corresponding to other known plant phytochrome structures. 8R44 is to our knowledge, the highest resolution structure for a plant phytochrome to date, providing novel details of the chromophore geometry and accurately defining the positions of all immobile non-hydrogen atoms. This is in particular important regarding water molecules, some of which play crucial roles in optimising photochemistry and protonation dynamics[33,34,42,43]. These structures thereby provide a sound basis for understanding the photochemistry of Pr in the $S_0$ quantum mechanical ground state. As in the case of phyB (PDB codes 6TBY and 6TC5)[9], the protein structures of the PCB and PΦB adducts are almost identical (Supplementary Figs. 3 and 4). This is surprising as, presumably, the rate of Pfr→Pr thermal reversion is restricted by the activation energy, yet the more rigid vinyl side chain would be expected to raise rather than lower the barrier. Apparent differences in water mobilities might provide a clue (see Supplementary Discussion).

We also present a 2.2 Å ambient temperature SFX structure of the PCB adduct (PDB code 9ER4). Comparison with an equivalent dataset collected independently revealed minimal differences (Supplementary Discussion and Supplementary Fig. 6), indicating that the data are reliable. However, 9ER4 deviates significantly from the cryogenic MX equivalent, 8R44, on account of thermal expansion. Taking this into account, local superimpositions then show the structures to be almost identical (Supplementary Fig. 8). Crystallographic X-ray damage in 8R44 is thus minimal, since SFX structures do not suffer from radiation artefacts. Detailed electron density and omit maps for 9ER4 are shown in Supplementary Figs. 11 and 12.

Our soybean phyA(nPAS-GAF) Pr structures are closely similar to the homologous region in cryo-EM structures of full-length *Arabidopsis* phyA[15–18] (Fig. 1c). This is remarkable given the sequence differences and the potential effects of domain interactions at various levels. Even the D273 residue critical for interaction with the tongue of the PHY domain is hardly changed in the nPAS-GAF structures despite the missing tongue interaction.

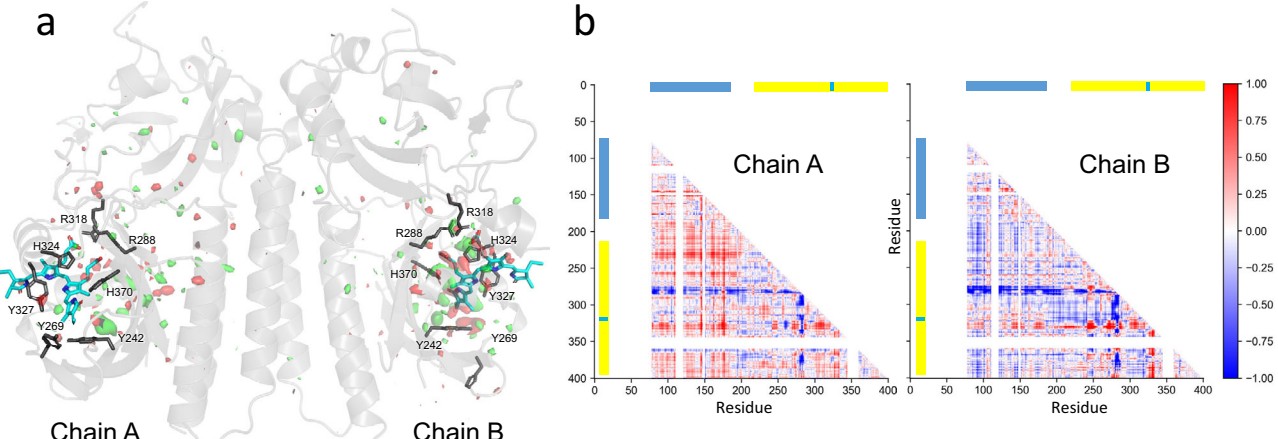

**Fig. 2 | Overall comparison of Pr and Pfr structures.** Pr and Pfr SFX ambient temperature structures (9ER4 and 9F4I, respectively). **a** Fo(light)-Fo(dark) electron density difference map contoured at 3.5 rmsd (red and green) with main chains A and B (light grey), pertinent side chains (dark grey) and chromophores (cyan) of 9ER4 superimposed. **b** Cα difference distance matrix comparisons (colour code, Å relative to 9ER4). nPAS & GAF domains, blue & yellow bars; Chromophore position,

cyan. The residues associated with the density differences are (chain A) C85, L157, Y168, A169, Y242, Y269, D273, L281, F282, M283, D285, Y327, V367, V368, C369, F376, R382, A384 & E386 and (chain B) A140, V156, F182, E189, M240, D248, Y264, H268, Y269, S278, L218, F282, M283, N285, T316, R318, S322, Y327, S336, V338, G365, H370 & F392.

## Pfr structures

Details of the chromophore region of the soybean phyA(nPAS-GAF) structure in the Pfr state (9F4I) are shown in Fig. 3a and b. Detailed electron density and omit maps are shown in Supplementary Figs. 11 and 12. The peptide chain is knotted around the nPAS domain, as in other phytochrome structures. The chromophore pocket is open to the medium (Fig. 3c and Supplementary Fig. 8), the cofactor itself showing periplanar ZZEssa geometry, as in all Pfr structures to date, indicating the expected $Z \rightarrow E$ photoisomerisation of the C15 = C16 double bond and D-ring photoflip (Fig. 4). The ca. 45 °C-D ring tilt in Pfr is similar to that in Pr. This angle might be significant since it primes the D-ring for isomerisation yet maintains π-electron coupling: the latter falls rapidly beyond 50° according to the cos² dependency in Hückel molecular orbital theory. Conversely, the A-ring tilt is almost unchanged. The cofactor as a whole shows a significant rotation in the direction of ring D, in accord with the flip-and-rotate model[44]. As the latter has in the meantime been verified also for various bacteriophytochromes (DrBphP[45–47] 4Q0J, 5C5K, 8AVX, 8AVV, 8AVW; IsPadC[48,49] 6ET7, 6SAX, 6SAW; XccBphP[50,51] 5AKP, 6PL0, 7L59, 7L5A; SaBphP2[52–54] 6BAO, 8UQI, 8UPK, 8UPH, 8UPM, 8UQK; PsBphP1[55] 8U4X, 8U62, 8U63, 8U64, 8U65) as well as for Arabidopsis phyB[16,19] (7RZW and 8YB4), we expect it to be valid generally. Indeed, the Arabidopsis phyB variant Y276H that mimics Pfr in transgenic plants, even in total darkness[56] shows a similar shift in chromophore position despite Pr-typical ZZZssa geometry (PDB 9IUZ)[19].

Bacteriophytochrome structures show the tilted D-ring to be above (α-facial of) the B-C ring plain in both Pr and Pfr. The D-ring is α-facial in all plant phytochrome structures too, including those for Pfr (9F4I (Fig. 4 and Supplementary Fig. 9a) and 8YB4[19] for phyA and phyB, respectively). This contradicts the proposal of Rockwell et al.[57] that, in contrast to bacteriophytochromes, the D-ring slumps to the β face in the Pfr state of phytobilin-based phytochromes such as phyA. That notion derived from CD spectra that show sign inversion in the red region upon plant phytochrome photoconversion, implying very different chiralities in Pr and Pfr, whereas inversion is not seen for bacteriophytochromes. The slumping effect was suggested to follow from an anti-clockwise photoflip of the D-ring in plant phytochromes due to steric interaction between the C13¹ and C17¹ methyl groups, whereas bacteriophytochromes would flip clockwise, avoiding methyl clashes. Borucki et al.[58] pointed out, however, that the conjugated π-orbital system includes the A-ring in biliverdin but not in phytobilins (PCB and

PΦB), the chromophores of bacteriophytochromes and plant phytochromes, respectively. Thereby, whereas A-ring movements might mask D-ring-associated chirality changes in bacteriophytochromes, this would not be expected in plant phytochromes. Indeed, genetic modification of the bacteriophytochrome Agp1 to allow adduction of the phytobilin PCB yields CD sign inversion[58]. In any case, our structures provide little evidence for A-ring movement, thus a clockwise flip of the D-ring would seem reasonable. Despite weak photochromicity and issues regarding multiphoton excitation, recent time-resolved pump-probe SFX measurements of SaBphP2 bacteriophytochrome PSM microcrystals confirm a clockwise flip of ring D during Pfr formation[59] (although similar studies of an nPAS-GAF construct implied an anticlockwise flip[60]). Taken together, the data imply that a clockwise D-ring photoflip and a rotation of the chromophore as a whole within the pocket are similar throughout the phytochrome family.

Other structural differences between the Pr (8R44 and 9ER4) and Pfr (9F4I) states of our phyA construct are concentrated around the chromophore (Figs. 5 and 6). Many of the observed shifts can be attributed to H-bonding changes associated with chromophore flip-and-rotation (Fig. 4 and Supplementary Fig. 9a) and are likely to be functionally significant, not least because similar changes are seen in phyB[16,19] and bacteriophytochromes[45,48–51,61]. We note that none of the residues referred to below show likely cryogenic artefacts (Supplementary Fig. 7). Particular differences between the Pr and Pfr structures are described below.

The tyrosine dyad Y242 and Y269 below (β-facial of) ring D undergoes radical rearrangement, both rings flipping over in concert with the D-ring (Y242 from t80° to m-30°, Y269 from m-85° to p90°[62]; Fig. 5a and Supplementary Fig. 9b), as also seen in Arabidopsis phyB[16,19] and generally in bacteriophytochromes (although, interestingly, not in the 4O0P and 4O01 Pr and Pfr + Pr mixed state structures of DrBphP[46]). The MAS NMR model for oat phyA3[33] agrees with such rotamer changes for Y242 but not Y269 (Y241 and Y268 in oat phyA3, respectively), the latter inconsistency arising from the assumption that the D-ring is β-facial in Pfr[57], a notion that now seems unlikely (see above). Resonance interactions between the side chain aromatic rings and the chromophore might be responsible for the bathochromic shift characteristic of Pfr formation. The Y242 residue and its homologues are particularly interesting. Several mutations at this site strongly inhibit R/FR photochromicity and induce strong fluorescence in plant

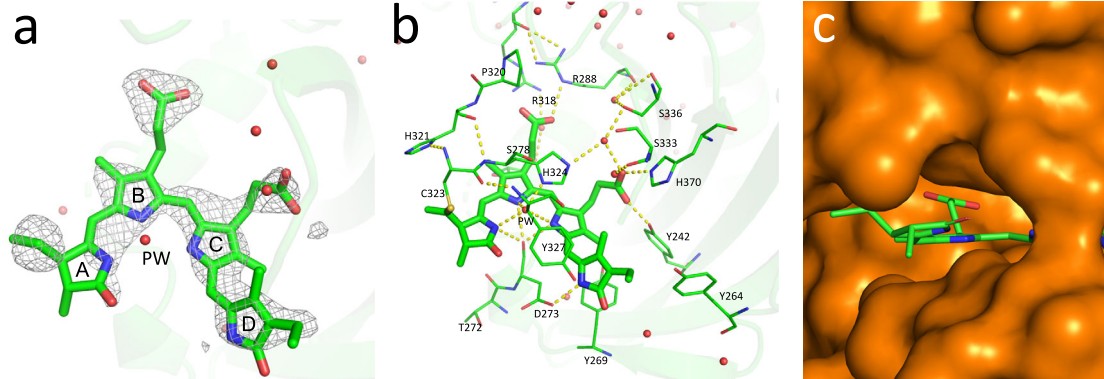

**Fig. 3 | Ambient temperature SFX structure of phyA(nPAS-GAF) as Pfr.** The 2.2 Å structure of the chromophore and pocket as Pfr derived from the mixed state SFX dataset (PDB code 9F4I, protomer B) is shown. Carbons, green. **a** q-weighted extrapolated electron density map of the chromophore contoured at 0.448 e⁻ Å⁻³ (1.39 rmsd), α-facial view. The ethyl side chain of ring D shows electron density both α- and β-facially. **b** amino acids and waters (red spheres) surrounding the chromophore, α-facial view. PW, pyrrole water. H-bonds are shown as dashed yellow lines. **c** GAF domain molecular surface (orange) showing the chromophore pocket open to the medium.

phytochromes[56,63] and Cph1[64,65] but not bacteriophytochromes: the 3D structure and function of Y176H, the homologous variant in Cph1, have recently been described in detail[66]. Moreover, in transgenic plants, homologous variants in phyB induce constitutively photomorphogenic seedling development, implying that the structures mimic the signalling properties of Pfr[56,63]. The cryo-EM structure of the *Arabidopsis* phyB Y276H variant (PDB 9IUZ) implies that this results from a rotational shift of the chromophore analogous to that seen in Pfr[19].

Interactions of the B-ring propionate (propB) with neighbouring R288 and R318 rearrange upon photoactivation (Fig. 5b and Supplementary Fig. 9a). The robust salt bridge between the carboxylate and R318, augmented by single H-bonds to the R288 guanidine and the H321 backbone nitrogen seen in Pr are replaced in Pfr by single H-bonds to the R318 guanidine and Nε of R288, qualitatively in agreement with MAS NMR data for oat phyA3[33]. In *Arabidopsis* phyB[19], the equivalent interactions are exclusively to R322, the R288 homologue, consistent with the accelerated Pfr-Pr thermal reversion in the R322Q variant[23]. Thus, the propB partner swap, first proposed in analogy to haem-based oxygen sensors[11], is less dramatic than in the bacteriophytochromes *Dr*BphP and *Xcc*BphP, more resembling that in *Is*PadC[48].

Although the calculated Pfr electron density map is ambiguous for propC, the latter seems to change its interactions radically in concert with chromophore movement within the pocket (Fig. 5b, c and Supplementary Fig. 9a). In Pr, the carboxyl group forms H-bonds to R288 and several water molecules, whereas in Pfr, H-bonds to H370 and Y242 are likely (see below). Similar changes are seen in bacteriophytochromes. The imidazole rings of H324 and H370 shift upon photoactivation (Fig. 5c, d and Supplementary Fig. 9c), as implied by MAS NMR for the equivalents in oat phyA3[33]. H324 probably buffers protonation of the B-ring nitrogen[42] through the polarising effect of propC in both Pr and Pfr, as the chromophore is known to be cationic in both parent states. Indeed, H324, the PW and the chromophore shift upwards (α-facially) and laterally by about 1 Å upon photoconversion (Figs. 4 and 5d). Protonation dynamics are important in photoactivation, however, the metaRc intermediate, representing a transient deprotonated state associated with proton release to the medium seen in prokaryotic phytochromes[67,68], and might act as a pawl in the photocycle[69]. The subtle H370 movement (Fig. 5b, c and Supplementary Fig. 9c) implies that a putative H-bond to the D-ring carbonyl in Pr is broken upon photoactivation. Although the angle for such a bond is unfavourable, when the residue is substituted as in wild-type *Sa*BphP1,

significant changes are seen even though R/FR photochromicity appears normal[54,70].

The ring positions of Y264, F282 and Y327 lining the chromophore pocket shift too (Fig. 5c and Supplementary Fig. 9c), although the H-bond between Y327 and the D273 carboxyl is retained in both parent states. The Y327-homologous Y263F variant in Cph1 shows inefficient photochemistry and enhanced fluorescence[71,72], whereas in *Dr*BphP it leads to Pfr-like structural features even in darkness[73].

## Cellular signalling

Intermolecular signalling of phyA Pfr in the plant cell is likely to result from more extensive structural movements than the side chain shifts described above. In order to mediate partner-dependent signalling, an exposed, superficial region of the protein must exhibit state-specific conformational changes to inhibit/activate the molecular interaction. Our phyA(nPAS-GAF) structures and others provide insight into possible signalling processes.

Remarkably, the phytochrome peptide chain forms a knot, the NTE passing through a loop of the GAF domain, thereby tying the nPAS and GAF domains together[10,74]. Nagatani proposed that the knot plays a central role in plant phytochrome signalling because phyB loss-of-signal mutants at R110, G111, G112 and R352 in *Arabidopsis* (K76, G77, K78 and R318 in soybean phyA) are associated with that structure[23,75,76]. Potentially, disruption of the H-bonding between propB and its partners might lead to backbone movements appropriate for signal output: indeed, liquid NMR data implied such changes in the F145S/L311E/L314E mutant of the DrBphP bacteriophytochrome photosensory module[77]. However, neither the Pfr structure presented here nor that of Wang et al. for phyB[19] shows appropriate shifts. We propose that, instead, the signalling defects in the *Arabidopsis* mutants arise from functional damage to the NTE rather than to the knot through which it passes. The primary role of the knot might be rather to suppress thermal mobility. MAS NMR studies of both Cph1 and oat phyA3 have shown that the molecule as a whole in Pr is much more mobile than in Pfr, comprising at least two sub-states with very different photoconversion quantum efficiencies[33-35,78-80]. Enhanced Pr mobility was also reported for *Dr*BphP and suggested to allow Pfr-like conformations to occur in darkness[81]. This would be particularly problematical in plants where Pfr is the signalling state and might additionally explain why phyA is rigorously excluded from the nucleus in darkness.

Unfortunately, because of its inherent mobility, none of the current plant phyA structures provides 3D information about the NTE,

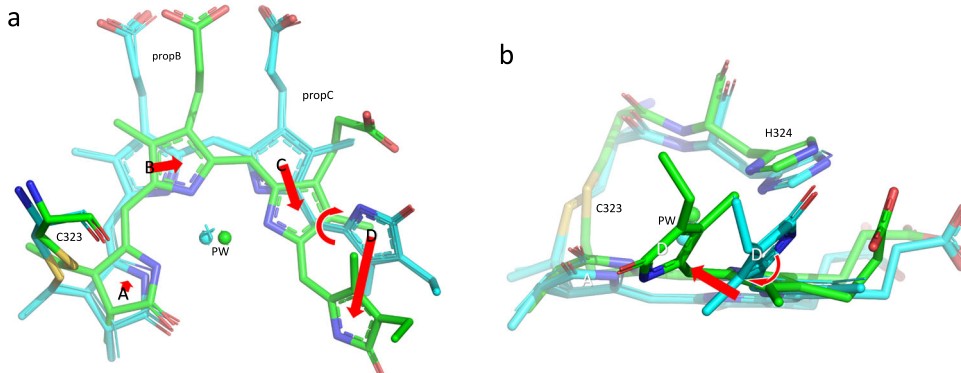

**Fig. 4 | Chromophore structures as Pr and Pfr.** Superimposed ambient temperature SFX Pr and Pfr (9ER4 and 9F4I, respectively; sticks) and cryogenic Pr (PDB 8R44, thin lines) structures. Carbons and waters, cyan and green in Pr and Pfr, respectively; oxygens, red; nitrogens, blue; waters, spheres and stars. Red arrows indicate the likely clockwise photoflip of ring D and the associated chromophore rotation within the pocket. The thioether attachment to C323 is shown along with H324 and the pyrrole water (PW). **a** view from the α face. **b** lateral view from ring D.

**Fig. 5 | Side chain and water shifts in Pr and Pfr.** Superimposed ambient temperature SFX Pr and Pfr (9ER4 and 9F4I, respectively; sticks) and cryogenic Pr (PDB 8R44, thin lines) structures showing state-dependent side chain shifts near the chromophore. Carbons and waters, cyan and green in Pr and Pfr, respectively; oxygens, red; nitrogens, blue; waters, spheres and stars; H-bonds, yellow dashes with distances in Å; PW, pyrrole water. **a** Tyrosine shifts below ring D. The rotamers are indicated. **b** Propionate side chain interactions with Arg, His and Tyr residues. **c** Further aromatic residue shifts near the chromophore. **d** H-bond interactions of chromophore nitrogens and propC with Asp and His residues and waters.

although MAS NMR of oat phyA implied that it is close to chromophore ring A, specifically in Pfr[33]. In *Arabidopsis* phyB, the NTE is extensive and is bound by ARR4, thereby inhibiting thermal Pfr→Pr reversion and thus increasing light sensititvity[82,83]. This interaction does not occur in phyA[83], but in both cases, the NTE is readily phosphorylated, reducing sensitivity[84]. The NTE in *At*phyB is poorly resolved in Pr[16], whereas in the Pfr-PIF6 complex[19] it is intimately associated with the active phyB binding domain (APB) of the PIF. The NTE of phyA might take part in similar interactions, but not only is it much shorter, no APB binding is seen. Only PIFs 1 and 3 carry the analogous active phyA binding domain

(APA) that is separate and quite different from the APB[85]. The APA binding site on phyA is still unknown.

The Q276-K286 helix, β-facial of propB and its contacts, lies at the surface of the antiparallel dimeric platform seen in cryo-EM structures of *Arabidopsis* phyA Pr and shifts towards the chromophore in Pfr (Fig. 6a, also Fig. 2). The homologous helices of bacteriophytochromes show similar state-dependent shifts, as do those of *Arabidopsis* phyB (Q310-R320) in the 8BY4 structure[19]. There, the helix of protomer B is partly covered by and interacts with the PIF6 hairpin that includes the APB motif. The S55-D64 helix and the V375-W397 hairpin of phyB Pfr, both likely to be mobile, leave a conspicuous cleft that might allow access for a further partner. Conversely, the corresponding A-chain helix lies exposed on the opposite side of the 8BY4 dimer.

The H321-D332 helix, α-facial of the chromophore and exposed at opposite ends of the phyA Pr antiparallel dimeric platform, also moves outwards in our Pfr structure (Fig. 6b, also Fig. 2). The homologous helices H355-G366 in *Arabidopsis* phyB Pfr[19] are exposed too. Although the helix might therefore be involved in signalling, there is no evidence for this, whereas the second histidine (H324 in our phyA structures) is conserved throughout the phytochrome superfamily and is important in chromophore protonation (Fig. 5c, d and Supplementary Fig. 9c; see above).

All cryo-EM structures of plant phytochromes, as Pr, show that the tongue of the PHY domain forms a lasso-like loop around the exposed side of the chromophore pocket (Fig. 1c), leaving it open to the medium. This is an important difference relative to prokaryotic phytochromes in which the tongue is compact, sealing the pocket at least in Pr and Pfr parent states. It is thus rather interesting that in protomer B of the 8YB4 phyB Pfr-PIF6 structure[19] the tongue is similarly compact, closing the chromophore pocket with the help of K56-Q109 of the NTE. The latter forms three α-helices showing hydrophobic interactions with chromophore ring A and the tip of an N-terminal antiparallel β-sheet of the PIF6 fragment. Conversely, nearly all of the NTE in protomer A is invisible, as in Pr, presumably on account of its mobility, and the PIF is not involved: thus, although the associated tongue is helical and compact, the chromophore pocket of protomer A is open to the medium.

The 9F4I and 8YB4 Pfr structures are surprisingly similar despite not only the sequence divergence but also the phyB-specific interaction with PIF6 in the case of 8YB4, which was considered to represent an induced fit[19]. Although all 8 PIFs in *Arabidopsis* bind phyB, only PIF1 and PIF3 bind phyA, both via a phyA-specific subdomain, the APA[85].

From the perspective of bacteriophytochromes, the results presented here provide a very limited description of the Pfr signalling process because only the nPAS and GAF domains are included. In particular, the adjoining PHY domain includes the tongue attached to the GAF domain. There, the radical tongue refolding seen upon photoconversion pulls on the PHY domain and thereby regulates the C-terminal "output module", usually an enzyme[46,86]. These assumed functions are not valid for plant phytochromes, however. As the cryo-EM structures of near full-length phyA and phyB in the Pr state show, the helix emerging from the PHY domain does not extend the helical spine into the C-terminal module as in prokaryotic phytochromes: in plant phytochromes, the helix connects to PAS1 via a flexible loop, rendering a mechanical effect unlikely. Furthermore, the PHY domain tongue of plant phytochrome Pr is quite differently structured from that of prokaryotic phytochromes: in all current plant phytochrome Pr structures, instead of completing the wall of the chromophore pocket adjacent to ring A as in prokaryotic representatives, it forms a lasso around an obvious pore, conspicuously allowing access to the surrounding medium[9] (Fig. 1c). Remarkably however, the pocket of protomer A is closed in the 8YB4 *Arabidopsis* phyB Pfr-PIF6 structure[19]: the tongue adopts a helix (as in bacteriophytochromes) and the lasso is lost, making room for helices of the NTE to interact with PIF6 and the GAF surface to close off the pocket. Conversely, in the A-chain

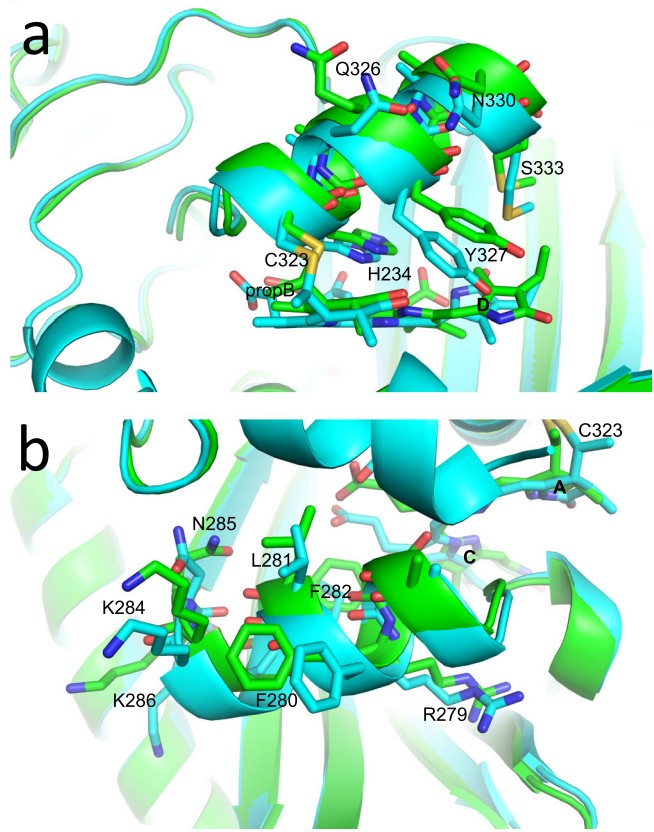

**Fig. 6 | Helix shifts in Pr and Pfr.** Superimposition of Pr (9ER4, cyan) and Pfr (9F4I, green) ambient temperature structures showing state-dependent shifts of helices and associated side chains near the chromophore. Carbons and waters, cyan and green in Pr and Pfr, respectively; oxygens, red; nitrogens, blue. **a** Helix H324-D332. **b** Helix Q276-K286.

protomer, although the tongue is helical, the NTE is invisible, probably because of its mobility, leaving the pocket wide open.

The PAS1 domain, known from early mutation studies to be important for signalling in both phyA and phyB[87], is not represented in any current structure on account of its mobility. A phyA Pr model including the AlphaFold2 prediction for PAS1 has been presented, however[7].

The near full-length Pr from cryo-EM studies[15–18] show nPAS, GAF, PHY and PAS-repeat forming an antiparallel (head-to-tail) platform upon which numerous partner molecules were expected to dock following more-or-less subtle structural shifts associated with Pfr. That assumption is questionable in the light of the 8YB4 and 9JLB structures[19,88] (published while the present paper was under review), describing the *At*phyB PSM as Pfr in complex with a fragment of one of its signalling partners, *At*PIF6. Reassuringly, the tongue is helical, and the chromophore is probably *ZZEssa* in both protomers. However, in contrast to Pr, they form a parallel (head-to-head) dimer, resembling that of bacteriophytochromes, but with a single PIF6 fragment bound to protomer B alone. Unfortunately, the PAS repeat and HKRD are missing from the structure, presumably as a result of their mobility. Also, the C-terminal dimerisation and DNA-binding domains of the PIF were not included in the construct. Thus, although the Wang et al. paper represents a milestone in plant phytochrome research, numerous questions remain. How does the interaction inhibit thermal Pfr→Pr reversion, and what is the structure of phyB Pfr without such interaction? Do other PIFs - in particular PIF3 - interact similarly? Can several PIFs bind simultaneously? Do they compete with each other? How does the interaction regulate PIF function? How does the PAS repeat contribute to Pfr function?

The latter raises an interesting point. The PIF family was identified by the binding of PIF3 with both phyA and phyB – however, with the C-terminal portion (PAS repeat and HKRD), not the PSM (as in the 8YB4 structure of the phyB-PIF6 fragment complex). How can that happen when PIF binding is Pfr-specific? A separate portion of the PIF might bind to the C-terminal portion of the phytochrome, and that might somehow be prevented in native Pr. PIFs are parallel dimeric, thus, the binding might be to a parallel-dimeric PAS repeat in the phytochrome, the configuration likely both in Pfr and of the isolated C-terminal portion, but not in Pr. That interaction might be associated specifically with photobody formation and PIF degradation. Note added in proof: The Choi laboratory has recently described important progress in this area[89].

## Photoactivation

The structural changes described here for phyA(nPAS-GAF) and elsewhere imply that most aspects of photoactivation in plant phytochromes accord with those in prokaryotic phytochromes. This is remarkable in view of (i) the large sequence divergence (about 35% identity based on 3D superimpositions of the photosensory modules), (ii) the very different domain organisations of plant and prokaryotic phytochromes, and not least (iii) our construct including only a small portion of the molecule. Although the similar behaviour enhances the value of research into photoactivation in prokaryotic phytochromes, whether the mechanism is universal remains to be proven. The present study provides little new information on the photoactivation process itself. Surprisingly, however, recent time-resolved SFX studies of bacteriophytochromes showed the disappearance of the PW only about 1 ps following photon absorption, well before D-ring isomerisation[60]. The significance of this is unclear, however, especially as the high actinic fluence rates used might have led to multi-photon excitation[90]. The photoflip itself results from electron density changes in the chromophore in the $S_1$ electronically excited state following photon absoption[79], but important open questions remain regarding the forces driving light-induced flip-and-rotation, the origin of the bathochromic shift characteristic of Pr→Pfr photoconversion, the chemistry of tongue refolding and, it would seem, the extensive domain restructuring implied by 8YB4 and 9JLB[19,88]. Specifically in the case of plant phytochrome signalling[87], the PAS repeat is important, yet its structure in the Pfr state remains unknown. 3D structures of phyA Pfr in complex with its partners, in particular the FHY1 transporter and PIF1 and PIF3, are eagerly awaited.

## Methods

### Sample preparation

Recombinant soybean (*Glycine max*) phyA(nPAS-GAF) holoproteins were produced in *E. coli* and purified essentially as described[9,91]. In the case of batch production of microcrystals, the procedure was adapted to a 5 litre Labfors II fermenter (Infors HT). The final sample buffer was 200 mM NaCl, 20 mM Hepes, pH 7.8, 5% (v/v) glycerol, 1 mM EDTA, 5 mM DTT.

### Single crystal production and X-ray diffraction

Single crystals for synchrotron crystallography were obtained by vapour diffusion in the hanging drop format, where protein solution at 15 mg/ml was mixed with the corresponding crystallisation solution (for PCB holoprotein: 0.3 M diethylene glycol, 0.3 M triethylene glycol, 0.3 M tetraethylene glycol, 0.3 M pentaethylene glycol, 0.1 M Tris (base), BICINE pH 8.5, 20% v/v PEG 500MME 10% w/v PEG 20000; for PΦB holoprotein: 0.01 M diethylene glycol, 0.01 M triethylene glycol, 0.01 M tetraethylene glycol, 0.01 M pentaethylene glycol, 0.1 M Tris (base), BICINE pH 9.0, 7% v/v MPD, 7% PEG 1000, 7% w/v PEG 3350) at 1:1 ratio and equilibrated against 500 µl of the reservoir at 283 K in darkness. Crystals were picked and flash-cooled in liquid nitrogen under a dim 490 nm safelight. X-ray diffraction data were collected at the BESSY II beamlines BL14.2 and BL14.3[92] using 100 K nitrogen and 50 K helium cryostreams for the PΦB (BL14.2, $\lambda = 0.918$ Å) and PCB (BL14.3, $\lambda = 0.896$ Å) holoprotein crystals, respectively. The 50 K cryostream was employed to investigate the effect of measurement temperature on the quality of the electron density map in disordered regions of the structure. The same crystal was also measured at 100 K without any noticeable effect on the map.

### Microcrystal production and analysis

Microcrystals were produced using a batch crystallisation protocol optimised with the help of the T-REXX endstation at EMBL (Hamburg). Crystal seed was produced using Seed Beads (Hampton) following the recommended protocol[93,94]. The seed slurry was mixed with the precipitant solution [0.1 M Tris (base), BICINE pH 8.5, 18% v/v PEG 500MME 9% w/v PEG 20000] to 2% v/v. Subsequently, equal volumes of the precipitant and holoprotein (20 mg/ml) solutions were mixed. Following brief vortexing, the solution was incubated at 293 K for up to 48 h in the dark. The volume of the gravity-settled microcrystals typically represented ~ 30% of the total volume.

UV-Vis spectra of the microcrystals were recorded with a Tecan Spark plate reader and black-walled microtitre plates. Microcrystals were spun down from the slurry at 1000 g for 10 min, and washed twice in protein-free precipitant solution. The suspension was then placed in a microtitre well and the plate spun at 1000 g for 10 min to sediment the microcrystals again. Spectra were recorded following saturating 730 nm and 650 nm LED irradiation to generate Pr and the Pr + Pfr photoequilibrium mixture, respectively. Finally, the microcrystals were dissolved in extraction buffer, and the spectra measured again.

The microcrystals were inspected by transmission electron microscopy (TEM). Briefly, holey carbon R1.2/1.3 quantifoil copper grids were glow discharged (Gloqube plus, Quorum Technologies) immediately before the addition of 3 µl of microcrystal suspension. After 30 s the suspension was blotted off and the adhering material washed twice with water droplets that were immediately removed. The crystals on the grid were then fixed by two short (1 s) incubations with 2% (w/v) uranyl acetate, followed by a 30 s incubation for the negative contrast staining. The adhering solution was blotted off and the grids were allowed to dry > 10 min prior to inspection. Electron micrographs were obtained using a Jeol TEM 2100-Plus with LaB6 cathode using 200 kV acceleration voltage. Electron diffraction was also recorded by inserting a selected area aperture with a 1 µm diameter into the electron beam path below the sample.

### Microcrystal delivery

Samples were delivered through DFFNs as described previously[95–97]. DFFNs were 3D printed (Photonic Professional GT, Nanoscribe) with sample liquid, sheath liquid and helium gas orifices of 75, 95 and 60 µm, respectively. The sheath liquid was ethanol with a flow rate of 20 µl/min. The sample flow rate was 20 µl/min from Shimadzu HPLC pumps (LC-20AD) using 100 µm inner diameter tubing. Photoactivation was achieved by irradiating the sample in line for 2.3 s at 25 W/m² from a 630 nm LED array, allowing 45 s for thermal relaxation prior to jetting. The helium flow was adjusted to give a final jet velocity of 40 m/s.

### SFX data collection and processing

The X-ray beam at the SPB/SFX instrument of the European XFEL (Schenefeld, Germany)[98] was tuned to a photon energy of 9.3 keV ($\lambda = 1.33$ Å) and an intratrain repetition rate of 564 kHz (185 pulses per train, 10 trains per second), with a pulse energy of 2 mJ, a pulse duration of 25 fs and an X-ray focal spot size of 3 µm. The X-ray intensity was attenuated with silicon foils, resulting in a pulse energy of 320 µJ impinging on the sample, also accounting for the transmission of 60% of the instrument at this configuration and photon energy.

Diffraction images were recorded with a 1 megapixel adaptive gain integrating pixel detector (AGIPD)[99] at 564 kHz separated into 5-min runs of *ca.* 600000 raw images each. Online monitoring of the experiment was provided by Karabo[100] and OnDA[101]. After automatic calibration of the images, they were indexed using xgandalf[102] and integrated with CrystFEL v0.10.2[103,104] via the EXtra-Xwiz pipeline[105] using options --highres=1.5, --peaks = peakfinder8, --min-snr = 5, --threshold = 70, --min-pix-count = 1, --max-pix-count = 100, --int-radius = 2,3,5, --local-bg-radius = 4, --max-res = 1200, --min-peaks = 0 and --multi for indexamajig.

For the 9ER4 Pr and 9F4I Pfr structures, data were acquired over a total of 60 and 65 min (6.1 and 6.2 million images), respectively. The overall indexing rates were 1.0% (dark, Pr) and 2.8% (LED irradiated, Pr +Pfr mixed state) (0.5% to 3.7% for individual five-minute runs). The data were merged with partialator (using options --max-adu = 65000, --iterations = 1, --model = unity), and the high-resolution cutoff was set at 2.2 Å based on reasonable values of SNR, Rsplit, CC1/2 and CC* in the outer resolution shell.

### Refinement

The Pr cryogenic macrocrystal structures from BESSY II, as well as the ambient temperature microcrystal structure from EuXFEL, were refined essentially as previously described using the CCP4 and PHENIX suites[9,106,107]. The structure was solved by molecular replacement using 6TC7 as the search model and refined using COOT 0.9.8.93[108]. The LED-irradiated dataset was scaled to the dark dataset using SCALEIT from the CCP4 package using Wilson scaling. Q-weighted[109] extrapolated structure factor amplitudes[41,110] were calculated for assumed occupancies ranging from 0.05 to 0.5 in small steps. Visual inspection of these maps indicated likely Pfr occupancies of ~0.11 and ~0.23 for the A- and B-chains, respectively. An initial Pfr model was built using the light-minus-dark and extrapolated maps, then combined with the Pr structure using the determined occupancies, with Pr as one conformer and Pfr as the other. The resulting complex model was then refined against the original red-irradiated dataset using PHENIX[107], allowing only the Pfr part of the structure to move. The scheme for deriving the Pfr map and structure is described in more detail in the Supplementary methods and Supplementary Fig. 10.

Molecular images were generated using PyMol 3, kindly provided by Schrödinger Corp.

### Data availability

Atomic coordinates and structure factors are publically available at wwPDB as follows: 8R44: Pr, PCB adduct, single crystal, from dark BESSY II synchrotron dataset collected at 50 K [https://doi.org/10.2210/pdb8R44/pdb]. 8R45: Pr, PΦB adduct, single crystal, from dark BESSY II synchrotron dataset collected at 100 K [https://doi.org/10.2210/pdb8R45/pdb]. 9ER4: Pr, PCB adduct, microcrystal slurry, from dark EuXFEL dataset collected at 294 K [https://doi.org/10.2210/pdb9ER4/pdb]. 9F4I: Pfr, PCB adduct, microcrystal slurry, determined from 630 nm irradiated EuXFEL dataset collected at 294 K (see Supplementary Information) [https://doi.org/10.2210/pdb9F4I/pdb]. The SFX diffraction images have been deposited in the CXIDB database as ID224 [https://doi.org/10.11577/2341297] and at https://doi.org/10.22003/XFEL.EU-DATA-004511-00.

### Code availability

The scripts used to derive the Pfr map and the difference distance plots have been deposited at GitHub and are available at https://doi.org/10.5281/zenodo.15213260.

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

## Acknowledgements
We thank the *Deutsche Forschungsgemeinschaft* (DFG) for support via SFB1078/B7 (J.H., S.N. and K.H.), Christina Lang, Tanja Gans and Sonja Graf (University of Giessen) for excellent technical assistance, and Prof. Arwen Pearson (University of Hamburg, Germany) for helpful discussions. We thank European XFEL (Schenefeld, Germany) for provision of X-ray free-electron laser beamtime at SPB/SFX under proposal numbers 4511 and 3376 (J.H. and K.H.), BESSY II (Helmholtz-Zentrum Berlin für Materialien und Energie, Germany) for provision of synchrotron beamtime under proposal number MX-221-00302 and others (J.H.), and the T-REXX endstation (P14.EH2; BMBF grant numbers 05K16GU1, 05K19GU1 and 05K22GU6) at PETRA-III (DESY, Hamburg, Germany) for initial microcrystal optimisation and thank the staff at each institution for their support.

## Author contributions
K.G., P-Y.C. and S.N. produced, purified and crystallised the holophytochrome samples. M.W. and C.F. organised the helium cryostream at BESSY. K.G. and P-Y.C. solved and refined the 8R44 and 8R45 structures. R.S., K.D., I.d.D., L.B., E.R., H.H., C.S. and J.S. prepared and characterised the microcrystal samples with J.H. and S.N. A.W. produced and tested the nozzles. K.D., R.d.W, J.B., K.K., C.K., J.C.P.K, M.K., F.H.M.K, A.S., T.S., J.V. and R.L. operated the SPB/SFX instrument and collected the SFX data with J.H., D.v.S., K.H. and S.N. D.v.S. carried out T-REXX tests and processed the SFX diffraction data. S.N. solved and refined the Pr and post-irradiation XFEL structures. D.v.S. and T.B. calculated the electron density difference maps. T.B. calculated the extrapolated Pfr structure factor amplitudes and the difference distance matrices. S.N. and J.H. interpreted the structures. J.H. wrote the manuscript with assistance from S.N. and in discussion with all authors. M.W., K.H., and J.H. devised and coordinated the project in discussion with A.M and R.B.

## Funding

## Competing interests
The authors declare no competing interests.

## Additional information
**Supplementary information** The online version contains Supplementary Mmaterial available at https://doi.org/10.1038/s41467-025-60738-w.

