## [Transparent Peer Review file · Nature Communications]

Pr and Pfr structures of plant phytochrome A

Corresponding Author: Professor Jon Hughes

Version 0:

Reviewer comments:

Reviewer #1

(Remarks to the Author)

Phytochromes (phy) are pivotal photoreceptors that govern physiological activities in a wide range of species, including plants, bacteria, and fungi. Plant phys, in particular, have been extensively studied for over 70 years. While certain aspects such as the photochemistry, spectral properties, signaling components, and physiological roles of plant phys have been well-studied in conjunction with their bacterial counterparts, there is a relative scarcity of information regarding the structures of plant phys. This hinders our understanding of the molecular mechanisms underlying plant phytochrome signaling pathways. In the past decade, several crystallography (Burgie et al., PNAS 2014; Nagano et al., Nature Plants 2020) and cryo-EM (Li et al., Nature 2022; Burgie et al., Nature Plants 2023; Zhang et al., Cell Research 2023; Wang et al., Cell Research 2023) studies have reported the structures of phyB or phyA, but only the inactive Pr structures have been elucidated. The structures of photoactivated Pfr and their signaling complexes are still lacking. This study focuses on the N-terminal nPAS-GAF module of soybean phyA, incorporated with PCB or PΦB phytybilin. Two canonical cryogenic crystal structures of Pr with extremely high resolutions (1.58 Å for the PCB version and 1.86 Å for the PΦB version) and two SFX crystal structures, including Pr and Pfr at ambient temperature, both with a resolution of 2.2 Å, were reported. The detailed structural analyses and comparisons in this study include three main highlights: (1) the depiction of most precise interactions between phytybilin and its binding GAF domain in the Pr structure of plant phyA, (2) the elucidation of conformational changes in phytybilin and its contacting residues of the GAF domain during photoconversion in a plant phytochrome, despite the use of PCB instead of PΦB, and these changes are conserved in bacterial phys, (3) the use of a newly-developed technique, SFX diffraction by XFEL, to solve the crystal structure at ambient temperature using microcrystals, which was confirmed by the classic cryogenic crystal structures solved by single crystals. The group has been pioneers in the structural study of bacterial and plant phytochromes, and are experts in the diverse biophysical and biochemical methods used for studying the photochemistry and photoconversion of the unique phytochrome photoreceptor family. Therefore, the structural analyses in this study are solid and accurate. Additionally, the authors demonstrate a thorough familiarity with the history of phytochrome research, resulting in a well-written paper with appropriately-cited references. Although there are no major concerns regarding this study, there are several minor issues that need to be addressed, as listed below, to improve the overall quality of the study.

Minor errors:

1. Fig. S1: Add one more "HKRD" for clarity.
2. Page 4, line 27 (P4, L27): "Fig. 1."
3. Fig. 1C: Add labels for domains such as nPAS and GAF, and motifs such as knot lasso and tongue protrusion of the PHY domain, as these claims have been mentioned in figure legends.
4. Fig. S3: Change the labels such as W518 to W-518 or something similar to distinguish the tryptophan residues. Additionally, clarify why these water molecules are labeled with numbers less than 400, such as W306 or W396. As the nPAS-GAF module of phyA should contain approximately 400 residues, these labels for water molecules may overlap with those for residues.
5. P7, L2: Provide the full name for "SPB." L5: Cite the supplementary discussion.
6. Fig. S5: Provide the full names for TEM and SAED.
7. Fig. S7: Delete "(Pr; 9ER4 vs. 8R44)" in the legend.
8. Fig. S8: Add the full name for "propB" in the legend.
9. P8, L8: Cite the supplementary discussion. L17: "Fig. 2." P9, L1: "Fig. 3." L9: "Fig. 1." P10, L2: "Fig. 3." L4: "Fig. 4."
10. P9, L13: Provide further description for "S0 ground state."
11. P10, L6: Further description for "cos2." L9: The recent cryo-EM structures of bacterial phytochromes can also be cited, including (Wahlgren et al., Nature Communications 2022; Burgie et al., Nature Communications 2024; Malla et al., Science

Advances 2024). L13: Provide the full name for "propC."

12. P11, L1: Refer to "Fig. 4," and remove the word "and." L6-L8: Rephrase this sentence and provide more specific description, such as "avoiding clashes between what?" L19: Provide the full name for "H-bond."

13. P12, L10: Correct Y248 to Y268.

14. P13, L6: Provide the full name for "metaRc." L9: Refer to "Fig. 5A."

15. Fig. 5D legend: Correct Y268 to Y269, and provide further description for "p, m, t" in the figures.

16. P15, L15 and P16, L20: Replace "HKL" with "HKRD" for consistency in the claim.

17. P16, L16-23: Cite a review paper or specific research articles to support these conclusions.

18. P17, L21-25: Pay closer attention to the parentheses. L27: Remove the word "at"

(Remarks on code availability)

Reviewer #2

(Remarks to the Author)

Phytochromes are a widespread family of red/far-red responsive photoreceptors in plants and micro-organisms, they utilize covalently attached bilin chromophores that enable photoconversion between red-absorbing (Pr) and far-red-absorbing (Pfr) forms.

In the present study, the authors determined the crystal structures of the N-terminal chromophore-binding module of soybean phytochrome A in both Pf and pfr states, and observed some conformational changes in the case of the chromophore rings and the corresponding residues. Though these changes parallel those in bacteriophytochromes, it provide a little information for the plant phytochromes considering their great sequence divergence and domain compositions.

Moreover, the structures presented in the paper only include the nPAS and GAF domain, the physiological signaling process for the Pfr, and the relative structural rearrangement in the presence of the C-terminal modules is also unclear.

Thus, the current research is more suitable for a specialized journal like Communication Biology or Structure.

(Remarks on code availability)

Reviewer #3

(Remarks to the Author)

I recommend that manuscript by Hughes and colleagues "Pr and Pfr structures of plant phytochrome A" be published after major revisions. Although noteworthy results are presented in the manuscript, several key issues have to be addressed:

- 1) including a more detailed explanation of how extrapolation maps were generated
- 2) detailed difference electron density maps in the vicinity of ring-D and neighboring amino acids that stabilize BV chromophore
- 3) detailed explanation of reasons for choosing clockwise rotation of D-ring
- 3) explanation of water molecule re-arrangement and/or lack thereof
- 3) updated references as some key references are missing, including relevant time-resolved crystallography experiments involving BphPs.

- The noteworthy results certainly include a first, detailed view of the Pfr state relevant to the plant phyA. The authors describe in detail all relevant structural changes and provide insight into key structural changes specific to plant Phys.
- The work will be of significance to the field and related fields. The work provides unique insight into Pfr structure of Plant Phytochrome. However, the PhyA used in this study is a truncated version lacking very important PHY domain that has been shown in the past to play very important role in phytochrome Pr/Pfr photoconversion. The work is original but it is missing some key references including time-resolved crystallography on BphPs and recent cryo-EM experiments highlighting a Pr/Pfr heterodimer of a bacterial BphP.

- Page 4 - lines 7-9 suggest that it is not clear why PHY domain is important for Pr/Pfr photoconversion. References below offer important insight regarding PHY domain is important during photoconversion.

1. Yang X, Stojković EA, Ozarowski WB, Kuk J, Davydova E, Moffat K. Light Signaling Mechanism of Two Tandem Bacteriophytochromes. *Structure*. 2015 Jul 7;23(7):1179-89. doi: 10.1016/j.str.2015.04.022.

2. Malla TN, Hernandez C, Muniyappan S, Menendez D, Bizhga D, Mendez JH, Schwander P, Stojković EA, Schmidt M. Photoreception and signaling in bacterial phytochrome revealed by single-particle cryo-EM. *Sci Adv*. 2024 Aug 9;10(32):eadq0653. doi: 10.1126/sciadv.adq0653.

3. Burgie ES, Basore K, Rau MJ, Summers B, Mickles AJ, Grigura V, Fitzpatrick JAJ, Vierstra RD. Signaling by a bacterial phytochrome histidine kinase involves a conformational cascade reorganizing the dimeric photoreceptor. *Nat Commun*. 2024 Aug 10;15(1):6853. doi: 10.1038/s41467-024-50412-y.

- The work in part supports the conclusions and claims. The additional evidence is needed as explained above, in particular on how the extrapolated maps were generated. The following reference needs to be included as it outlines detailed description of D-ring rotation, amino acid and water network rearrangement that differs from PAS-GAF construct described in Claesson et al, *eLife* 2020.

4. Carrillo M, Pandey S, Sanchez J, Noda M, Poudyal I, Aldama L, Malla TN, Claesson E, Wahlgren WY, Feliz D, Šrajter V,

Maj M, Castillon L, Iwata S, Nango E, Tanaka R, Tanaka T, Fangjia L, Tono K, Owada S, Westenhoff S, Stojković EA, Schmidt M. High-resolution crystal structures of transient intermediates in the phytochrome photocycle. *Structure*. 2021 Jul 1;29(7):743-754.e4. doi: 10.1016/j.str.2021.03.004.

- It seems that there are no flaws in the data analysis, interpretation and / or conclusions. However, revision is required to further support direction of D-ring rotation and movements of key amino acids and protein backbone following photoexcitation.
- With more detailed description in methodology and accounting for references mentioned here, the revised work should meet the expected standards in phytochrome field.
- Visual representation of Figures 4-6 would benefit from different choices of color for BV chromophore in the Pr and Pfr states - blue and green when superimposed are rather similar colors. It would be better to use more contrasting colors - such as blue and orange for example...

(Remarks on code availability)

Version 1:

Reviewer comments:

Reviewer #1

(Remarks to the Author)

The authors have addressed most, though not all, of my questions. I have no specific questions regarding the conclusions of this revised paper. However, the newly added notes concerning the recently published phyB-Pfr-PIF6 structure contain some inaccurate and misleading content. For instance, in the recent *Cell* paper, PIF6 binding was shown to substantially inhibit the thermal reversion of phyB, a conclusion strongly supported by structural observations. The authors, however, seem to connect this conclusion to the argument against phyB functioning as a thermal sensor. It is important to clarify that it is phyB, not the phyB-PIF6 complex, that acts as a thermal sensor. In planta, thermal sensing and PIF binding of phyB are likely two distinct events. Therefore, I suggest the authors delete this sentence in discussion.

(Remarks on code availability)

Reviewer #3

(Remarks to the Author)

(Remarks on code availability)

Version 2:

Reviewer comments:

Reviewer #1

(Remarks to the Author)

The current version of the manuscript is well-organized, particularly the major sections, including results, discussion, main figures, and supplementary figures. I have no major concerns, but there are still literal errors that the authors should thoroughly check throughout the manuscript. For instance:

1. On page 7, line 9, and in the legends of Figure 2a, the term "Fo-Fo" might confuse readers, even those with expertise in crystallography. It would be more precise to use the description in the legends of Figure S9, such as Fo(light)-Fo(dark) or Fo(Pfr)-Fo(Pr).
2. On page 21, line 10, the abbreviation HKLD is never used in the main text; instead, HKRD is used. Replace it with HKRD.

(Remarks on code availability)

Authors' replies to reviewers' comments (the page and line numbers given by the reviewers refer to the original submission).

Reviewer #1 (Remarks to the Author):

Phytochromes (phy) are pivotal photoreceptors that govern physiological activities in a wide range of species, including plants, bacteria, and fungi. Plant phys, in particular, have been extensively studied for over 70 years. While certain aspects such as the photochemistry, spectral properties, signaling components, and physiological roles of plant phys have been well-studied in conjunction with their bacterial counterparts, there is a relative scarcity of information regarding the structures of plant phys. This hinders our understanding of the molecular mechanisms underlying plant phytochrome signaling pathways. In the past decade, several crystallography (Burgie et al., PNAS 2014; Nagano et al., Nature Plants 2020) and cryo-EM (Li et al., Nature 2022; Burgie et al., Nature Plants 2023; Zhang et al., Cell Research 2023; Wang et al., Cell Research 2023) studies have reported the structures of phyB or phyA, but only the inactive Pr structures have been elucidated. The structures of photoactivated Pfr and their signaling complexes are still lacking. This study focuses on the N-terminal nPAS-GAF module of soybean phyA, incorporated with PCB or PΦB phytobilin. Two canonical cryogenic crystal structures of Pr with extremely high resolutions (1.58 Å for the PCB version and 1.86 Å for the PΦB version) and two SFX crystal structures, including Pr and Pfr at ambient temperature, both with a resolution of 2.2 Å, were reported. The detailed structural analyses and comparisons in this study include three main highlights: (1) the depiction of most precise interactions between phytobilin and its binding GAF domain in the Pr structure of plant phyA, (2) the elucidation of conformational changes in phytobilin and its contacting residues of the GAF domain during photoconversion in a plant phytochrome, despite the use of PCB instead of PΦB, and these changes are conserved in bacterial phys, (3) the use of a newly-developed technique, SFX diffraction by XFEL, to solve the crystal structure at ambient temperature using microcrystals, which was confirmed by the classic cryogenic crystal structures solved by single crystals. The group has been pioneers in the structural study of bacterial and plant phytochromes, and are experts in the diverse biophysical and biochemical methods used for studying the photochemistry and photoconversion of the unique phytochrome photoreceptor family. Therefore, the structural analyses in this study are solid and accurate. Additionally, the authors demonstrate a thorough familiarity with the history of phytochrome research, resulting in a well-written paper with appropriately-cited references. Although there are no major concerns regarding this study, there are several minor issues that need to be addressed, as listed below, to improve the overall quality of the study.

We thank the reviewer for his/her positive comments on our paper and in particular for the exceedingly thorough editing job!

Minor errors:

1. Fig. S1: Add one more "HKRD" for clarity.

Done. The experimental construct has been added.

2. Page 4, line 27 (P4, L27): "Fig. 1."

Done.

3. Fig. 1C: Add labels for domains such as nPAS and GAF, and motifs such as knot lasso and tongue protrusion of the PHY domain, as these claims have been mentioned in figure legends.

Done – however, the GAF domain predominates in this view such that it cannot be labelled usefully. Instead, the colour ranges have been added to the legend.

4. Fig. S3: Change the labels such as W518 to W-518 or something similar to distinguish the tryptophan residues.

Done. The labels are now "Wat000". Two further waters unsubstantiated in 8R45 have been added.

Additionally, clarify why these water molecules are labeled with numbers less than 400, such as W306 or W396. As the nPAS-GAF module of phyA should contain approximately 400 residues, these labels for water molecules may overlap with those for residues.

The water numbering in the submitted structures is independent of that for the amino acids.

5. P7, L2: Provide the full name for "SPB." L5: Cite the supplementary discussion.

Done.

6. Fig. S5: Provide the full names for TEM and SAED.

Done.

7. Fig. S7: Delete "(Pr; 9ER4 vs. 8R44)" in the legend.

Done.

8. Fig. S8: Add the full name for "propB" in the legend.

Done.

9. P8, L8: Cite the supplementary discussion. L17: "Fig. 2." P9, L1: "Fig. 3." L9: "Fig. 1." P10, L2: "Fig. 3." L4: "Fig. 4."

Done.

10. P9, L13: Provide further description for "S0 ground state."

It now reads "in S_0 , the quantum mechanical ground state" to clarify the meaning of S_0 - or was it meant that we should describe why a high resolution structure is valuable in understanding molecular function?

11. P10, L6: Further description for "cos2." L9: The recent cryo-EM structures of bacterial phytochromes can also be cited, including (Wahlgren et al., Nature Communications 2022; Burgie et al., Nature Communications 2024; Malla et al., Science Advances 2024). L13: Provide the full name for "propC."

Done.

12. P11, L1: Refer to "Fig. 4," and remove the word "and."

Done.

L6-L8: Rephrase this sentence and provide more specific description, such as "avoiding clashes between what?"

Done. Reviewer #3 also requested additions to this section of the MS (see below). We hope that the modified paragraph now meets with approval.

L19: Provide the full name for "H-bond."

Isn't "H-bond" a standard abbreviation? If not, we would of course be happy to define it in the main text. We have added it to the list of abbreviations, just in case....

13. P12, L10: Correct Y248 to Y268.

Done.

14. P13, L6: Provide the full name for "metaRc."

Isn't that the full name?

L9: Refer to "Fig. 5A."

Done.

15. Fig. 5D legend: Correct Y268 to Y269, and provide further description for "p, m, t" in the figures.

Done. Lovell et al. 2000 is now cited in the main text and figure legend.

16. P15, L15 and P16, L20: Replace "HKL" with "HKRD" for consistency in the claim.

Done.

17. P16, L16-23: Cite a review paper or specific research articles to support these conclusions.

The key papers are now cited.

18. P17, L21-25: Pay closer attention to the parentheses. L27: Remove the word "at"

Done.

We hope that the changes we have made meet with the reviewer's approval and that he/she can now recommend publication.

Reviewer #2 (Remarks to the Author):

Phytochromes are a widespread family of red/far-red responsive photoreceptors in plants and micro-organisms, they utilize covalently attached bilin chromophores that enable photoconversion between red-absorbing (Pr) and far-red-absorbing (Pfr) forms.

In the present study, the authors determined the crystal structures of the N-terminal chromophore-binding module of soybean phytochrome A in both Pf and pfr states, and observed some conformational changes in the case of the chromophore rings and the corresponding residues. Though these changes parallel those in bacteriophytochromes, it provides a little information for the plant phytochromes considering their great sequence divergence and domain compositions.

Moreover, the structures presented in the paper only include the nPAS and GAF domain, the physiological signaling process for the Pfr, and the relative structural rearrangement in the presence of the C-terminal modules is also unclear.

Thus, the current research is more suitable for a specialized journal like *Communication Biology* or *Structure*.

We are sorry that this reviewer considers our work to be of little value. We point out, however, that at the time of submission it includes the first 3D structural information for any plant phytochrome in the Pfr signalling state. This is particularly important in relation to the distantly related prokaryotic phytochromes that signal via Pr-dependent activity of C-terminal enzyme modules, whereas in plant phytochromes the signal arises largely (though not exclusively) from the N-terminal photosensory module (see Hughes & Winkler (2024) *Ann Rev Plant Biol*) that is the focus of our work. In our view, the structural similarities to bacteriophytochromes are all the more remarkable considering the great sequence divergence. Our work is additionally novel for Pr in providing 3D structural information at unprecedented resolution, in quantifying the effects of cryogenic vs. ambient temperature during structural data collection, as well as in describing the surprisingly small structural differences between PCB and PΦB adducts.

We have added a final paragraph describing the phyB-PIF interaction study of Wang *et al.* (2024) that was published while our MS was still under review. That paper describes the cryoEM structure of a PIF6 fragment bound to a parallel-dimeric photosensory module in the Pfr state. We acknowledge that the work represents an important additional step forward despite the rather low resolution of the structures. Regarding the reviewer's comment on the role of the C-terminal region, we note that in Wang *et al.* too, the role of the C-terminal module remains unclear beyond that of dimerization propensity. In comparing the papers, it is also important to recognize the extensive functional differences between phyB and phyA, including but by no means restricted to the separate binding sites used by PIFs in each case.

Reviewer #3 (Remarks to the Author):

I recommend that manuscript by Hughes and colleagues "Pr and Pfr structures of plant phytochrome A" be published after major revisions. Although noteworthy results are presented in the manuscript, several key issues have to be addressed:

1) including a more detailed explanation of how extrapolation maps were generated

An extensive description of the method has been added to the SI.

2) detailed difference electron density maps in the vicinity of ring-D and neighboring amino acids that stabilize BV chromophore.

The chromophore is PCB, not BV. Two such difference maps are included in the SI (Fig. S9).

3) detailed explanation of reasons for choosing clockwise rotation of D-ring.

As requested by reviewer #1, we have rewritten the discussion at this point to clarify the argument and added information from recent time-resolved SFX work.

3) explanation of water molecule re-arrangement and/or lack thereof

We assume that the reviewer is referring to water movements associated with Pr→Pfr photoconversion. Although at $\sim 1.6 \text{ \AA}$ the 8R44 structure probably resolves all immobile water molecules in Pr, this is not the case for the $\sim 2.1 \text{ \AA}$ SFX structures. The e^- density map provides evidence that the PW shifts by $\sim 1 \text{ \AA}$ relative to the chromophore, other movements of clearly identifiable waters near the chromophore are much smaller. However, in a few cases the SFX map in particular for Pfr is not sufficiently clear to identify a water and its potential movement.

3) updated references as some key references are missing, including relevant time-resolved crystallography experiments involving BphPs.

As our paper does not include any data on structural kinetics, we had decided to save space by minimising this topic. We agree with the reviewer, however, that the time-resolved SFX work is important and have thus expanded the discussion in particular regarding the D-ring photoflip. This recent work is of course very relevant to that discussion and thus should have been included from the start. We are therefore grateful for the reviewer's suggestion.

- The noteworthy results certainly include a first, detailed view of the Pfr state relevant to the plant phyA. The authors describe in detail all relevant structural changes and provide insight into key structural changes specific to plant Phys.
- The work will be of significance to the field and related fields. The work provides unique insight into Pfr structure of Plant Phytochrome.

We thank the reviewer for these positive comments.

However, the PhyA used in this study is a truncated version lacking very important PHY domain that has been shown in the past to play very important role in phytochrome Pr/Pfr photoconversion.

We do not doubt that the PHY domain is important – and indeed it is missing from the construct studied in this paper – but point out that, whereas phytochromes generally require the PHY domain to stabilize Pfr, plant phyA is all but unique in showing almost normal photoreversibility even when everything C-terminal of the GAF domain is deleted. Furthermore, Viczian et al. (2012) showed that the N-terminal 406-residue fragment of *Arabidopsis* phyA is physiologically active when dimerised in the nucleus. We also take heart from the great structural similarity of our nPAS-GAF Pfr structure to that of 8YB4 recently published.

The work is original but it is missing some key references including time-resolved crystallography on BphPs and recent cryo-EM experiments highlighting a Pr/Pfr heterodimer of a bacterial BphP.

We hope to have resolved most of these issues by expanding the discussion to include additional studies – while retaining the primary focus on plant phytochromes.

- Page 4 - lines 7-9 suggest that it is not clear why PHY domain is important for Pr/Pfr photoconversion. References below offer important insight regarding PHY domain is important during photoconversion.

1. Yang X, Stojković EA, Ozarowski WB, Kuk J, Davydova E, Moffat K. Light Signaling Mechanism of Two Tandem Bacteriophytochromes. *Structure*. 2015 Jul 7;23(7):1179-89. doi: 10.1016/j.str.2015.04.022.

2. Malla TN, Hernandez C, Muniyappan S, Menendez D, Bizhga D, Mendez JH, Schwander P, Stojković EA, Schmidt M. Photoreception and signaling in bacterial phytochrome revealed by single-particle cryo-EM. *Sci Adv*. 2024 Aug 9;10(32):eadq0653. doi: 10.1126/sciadv.adq0653.

3. Burgie ES, Basore K, Rau MJ, Summers B, Mickles AJ, Grigura V, Fitzpatrick JAJ, Vierstra RD. Signaling by a bacterial phytochrome histidine kinase involves a conformational cascade reorganizing the dimeric photoreceptor. *Nat Commun*. 2024 Aug 10;15(1):6853. doi: 10.1038/s41467-024-50412-y.

With due respect to the reviewer, we think that a misunderstanding has arisen. We were specifically querying what feature/s of the PHY domain is/are responsible for Pfr stabilization, rather than the role of the PHY domain in regulating bacteriophytochrome signalling, the primary focus of the papers listed by the reviewer. Our MS is focused on plant phytochromes that in any case signal differently. The signalling functions of the PHY domain in bacteriophytochromes are therefore scarcely relevant. The tongue of the PHY domain refolds from β -sheet in Pr to α -helix in Pfr, a process that is crucial in photoconversion in most phytochromes. But how does the refolding proceed? Is the refolding process itself perhaps significant in photoproduct formation? Whatever these actions of the tongue might be, it seems that phyA is unusual in being able to photoconvert without them. Unfortunately, several static 3D structures of phyA and phyB with and without the PHY domain as Pr and Pfr, whose comparison with available data might provide valuable clues, are still missing. Serial SFX data would be even better, but the crystals must allow photoconversion to Pfr. Such experiments with our phyA(PG) construct are scheduled for 05.2025.

- The work in part supports the conclusions and claims. The additional evidence is needed as explained above, in particular on how the extrapolated maps were generated.

See above.

The following reference needs to be included as it outlines detailed description of D-ring rotation, amino acid and water network rearrangement that differs from PAS-GAF construct described in Claesson et al, eLife 2020.

4. Carrillo M, Pandey S, Sanchez J, Noda M, Poudyal I, Aldama L, Malla TN, Claesson E, Wahlgren WY, Feliz D, Šrajcar V, Maj M, Castillon L, Iwata S, Nango E, Tanaka R, Tanaka T, Fangjia L, Tono K, Owada S, Westenhoff S, Stojković EA, Schmidt M. High-resolution crystal structures of transient intermediates in the phytochrome photocycle. *Structure*. 2021 Jul 1;29(7):743-754.e4. doi: 10.1016/j.str.2021.03.004.

As we present no data that defines the direction of rotation directly, we did not want to expand the MS extensively regarding that issue. However, having raised the topic in relation to the disposition of the D-ring, we have added a brief consideration of the TR-SFX papers, as requested. We have also modified the paragraph following the comments of reviewer #1.

- It seems that there are no flaws in the data analysis, interpretation and / or conclusions. However, revision is required to further support direction of D-ring rotation and movements of key amino acids and protein backbone following photoexcitation.

We thank the reviewer for these positive comments on our analyses. Again, we do not present any data that directly show the direction of D-ring rotation, but rather suggest - on the basis of Rockwell et al. (2009) - that if the D-ring remains α -facial in Pfr, the direction of rotation would likely be clockwise, avoiding a clash between the C13¹ and C17¹ methyls. We have modified the relevant text as requested by reviewer #1 to make this clearer and added the more recent TR-SFX findings too (see above).

Regarding the movements of certain amino acid side chains and portions of the backbone, we make no statement beyond what we observe from superimpositions of Pr and Pfr structures.

- With more detailed description in methodology and accounting for references mentioned here, the revised work should meet the expected standards in phytochrome field.

We thank the reviewer for his/her helpful comments and hope that the revised MS now meets with approval.

- Visual representation of Figures 4-6 would benefit from different choices of color for BV chromophore in the Pr and Pfr states - blue and green when superimposed are rather similar colors. It would be better to use more contrasting colors - such as blue and orange for example...

The chromophore is PCB (or P Φ B), not BV, and the colours used are cyan and green for Pr and Pfr, respectively. We chose those colours to match the colour of light transmitted through phytochrome solutions in each state. We recognise, however, that the issue raised by the reviewer is significant. We have not changed this scheme in the revised MS but will ensure that the final published version will show an appropriately clear distinction between the two states.

Other revisions (marked in .docx file):

1. Numerous trivial corrections to text and figures have been made.
2. The abstract has been condensed to 150 words.
3. The figure legend style has been updated. The legends have been placed together in the main text file.
4. Embedded figures in the main text file have been removed and placed in separate files.
5. Fig. 3c and comment on chromophore access to medium has been added.
6. "Table 1" detailing the crystallographic statistics has been moved to the SI.
7. Several references have been added.
8. A comment on mobility of Pfr N-terminus has been added.
9. A comment that the chromophore pocket is open to the medium has been added
10. Following editorial agreement, a final paragraph discussing Wang et al. (2024) has been added.

Authors' reply to reviewers' comments

Reviewer #1

The authors have addressed most, though not all, of my questions. I have no specific questions regarding the conclusions of this revised paper. However, the newly added notes concerning the recently published phyB-Pfr-PIF6 structure contain some inaccurate and misleading content. For instance, in the recent Cell paper, PIF6 binding was shown to substantially inhibit the thermal reversion of phyB, a conclusion strongly supported by structural observations. The authors, however, seem to connect this conclusion to the argument against phyB functioning as a thermal sensor. It is important to clarify that it is phyB, not the phyB-PIF6 complex, that acts as a thermal sensor. In planta, thermal sensing and PIF binding of phyB are likely two distinct events. Therefore, I suggest the authors delete this sentence in discussion.

We are sorry if we failed to answer some earlier questions (and would still be happy to do so). As suggested by the reviewer, we have removed thermosensing from our discussion of the Wang *et al.* paper. If there is other "inaccurate and misleading content", it would be helpful to know what is meant so that we can correct it. We take the opportunity to thank the reviewer for his/her exacting corrections and otherwise enthusiastic comments on our work.

Reviewer #3 provided comments to the editors only. These were summarized as follows:-

* That the Pfr structure used for interpretation of photoconversion of phyA from Pr to Pfr is based on the red-irradiated (Pr + Pfr mixed state) microcrystals and thought it would be necessary to provide a more detailed interpretation of the difference electron density map in the chromophore binding pocket.

We thank the reviewer for giving us the opportunity to expand the description of the method in the SI: that now includes the section "Construction of the structural models" describing in detail specific features of the Pr and Pfr models in relation to the corresponding maps. This is also now illustrated extensively in Fig. S9 (see below).

* That Figure S9 is not sufficient in particular with regard to D-ring rotation and thought that additional justification was needed for the interpretation of the D-ring rotation as well as structural changes in other parts of the PCB chromophore.

We have addressed this criticism, firstly, by replacing the original Fig. S9 with several large illustrations of the of the chromophore region showing e^- density and the associated refined structures, and by adding the "Construction of the structural models" section to the SI text, as mentioned above. Secondly, we have improved Fig. 2a to provide a clearer view of the position and extent of the changes that follow photoexcitation of Pr. Thirdly, we have added statements in the main text (page 7, lines 20 to 21 and page 10, line 17) pointing out that propC of the chromophore is poorly resolved in the Pfr structure.

* That additional justification was needed to account for significance of PHY domain in light of

Wang *et al* 2024 showing critical conformational changes in the PHY domain of phyB. Reviewer #3 believed that the findings of Wang *et al.* 2024 should be integrated earlier into the manuscript.

We now include references to the finding of Wang *et al.* at several points in the MS, namely the Pr/Pfr domain rearrangement (page 3, lines 27 to 31), the cofactor shift (page 10, line 31 to page 11, line 4), and the lack of knot involvement (page 12, line 15). We have deleted the earlier "appendix" and added an extensive description of the results and significance of Wang *et al.* in the section "Structure of phyB as Pfr" (pages 14 and 15) of the Discussion, highlighting the surprisingly large difference between the phyB domain arrangements in Pr and Pfr, and posing some questions that remain unanswered. We also point out that phyA and phyB are associated with systems that have very different molecular properties and functions: in particular, whereas all PIFs bind phyB, only two also possess the separate binding domain for phyA – and PIF6 isn't one of them. Thus, we question whether the Pfr states of phyA and phyB in complex with different partners are "the same", despite the fact that our Pfr structure shows close similarities to that of phyB in Wang *et al.* (page 15, lines 11 to 16).

We have not otherwise expanded the structural comparisons (pages 10 and 11) to include details from Wang *et al.* Although we would be delighted to do this (it would certainly increase the value of the paper), we considered that each illustration comparing Pr and Pfr in phyA (Figs. 4a & b, 5a-d, 6a & b) would have to be accompanied by an equivalent for phyB (given that 4-fold superimpositions would be almost useless). We will make these changes upon request if the reviewer and editor agree.

Despite the reviewer's comment regarding the significance of the PHY domain, we have not opened this theme in discussing Wang *et al.*: our paper is only relevant to PHY domain function through its absence, as discussed in the Introduction regarding photochromicity (page 4, lines 14-18). We have, however, rewritten part of the Discussion to leave more room for PHY domain function in plant phytochromes (between pages 13 and 14). We would add the PHY domain to our discussion of Wang *et al.* if requested, but point out that a connection to our work on phyA would be entirely speculative.

We have also asked Reviewer #1 to provide feedback on Reviewer #3's comments. Reviewer #1 commented that they agree with Reviewer #3 that the difference maps as currently provided are insufficient to support the major conclusion that the authors have successfully obtained the Pr and Pfr structures of plant phyA.

We hope that Reviewer #1 too will now be satisfied with our submission, in particular through the new Fig. S9.

Authors' reply to reviewers

Reviewer #1

Remarks to the Author:

The current version of the manuscript is well-organized, particularly the major sections, including results, discussion, main figures, and supplementary figures. I have no major concerns, but there are still literal errors that the authors should thoroughly check throughout the manuscript. For instance:

- 1. On page 7, line 9, and in the legends of Figure 2a, the term "Fo-Fo" might confuse readers, even those with expertise in crystallography. It would be more precise to use the description in the legends of Figure S9, such as Fo(light)-Fo(dark) or Fo(Pfr)-Fo(Pr).*

- 2. On page 21, line 10, the abbreviation HKLD is never used in the main text; instead, HKRD is used. Replace it with HKRD.*

Thank you once again for your continuing positive comments and valuable suggestions. We have made the specific changes you suggest. In adding comparisons with the phyB Pfr structure of Wang *et al.* (2024) as had been requested by reviewer #3, we have streamlined part of the Discussion so that it now proceeds more logically from the N- to the C-terminus while nevertheless considering domain interactions. We hope that you will approve of these changes even though you liked the earlier version!

The changes are as follows (the numbering is for the January submission without markups, unless otherwise stated):

- p9 lines 22 to 25: not necessary, therefore deleted.
- p10 lines 08-23: moved to p9 line 25, thereby allowing p10 lines 24 to p11 line 02 to adjoin p10 line 07.
- p11 lines 07 to 09 and lines 16 to 24: not necessary, therefore deleted.
- p12 lines 19 to 27: extensively expanded with material from Wang *et al.* whereby the section "Structure of phyB as Pfr" (p14 lines 01 to 29) has been deleted.
- p13 lines 10 to 19: already mentioned in the Introduction and partly included in the new paragraph discussing PIF binding (p 14, lines 01 to 08 of the new submission), therefore deleted.
- p13 lines 20 to 28: moved to p12 line 19.

Reviewer #3

Remarks to the Author:

(none)

We understand that you might not have had time to process the January submission, especially given your earlier extensive and helpful comments and suggestions. We had agreed in principle with the idea of including comparisons with Wang *et al.* (2024) but considered that further illustrations might be necessary. In the current version, we have indeed integrated the Wang *et al.* findings but without expanding the figures, and at the same time streamlined the Discussion (see listed changes above). We hope that the MS thereby corresponds much more closely to your original suggestion. We acknowledge that these changes should have been made earlier but hope that you will be able to comment positively on the new submission.